# Engineering dimer mutants of human geranylgeranyl pyrophosphate synthase

**Sean J. Ezekiel**[ID]**, Mackenzie Searle, Jaeok Park**[ID]*

Department of Biochemistry, Memorial University of Newfoundland, St. John's, Newfoundland and Labrador, Canada

* jaeok.park@mun.ca

**Data Availability Statement:** All crystallographic data files for this study are publicly available from the Worldwide Protein Data Bank repository (https://doi.org/10.2210/pdb9csl/pdb). All other

## Abstract

Geranylgeranyl pyrophosphate synthase (GGPPS), a key enzyme in protein prenylation, plays a critical role in cellular signal transduction and is a promising target for cancer therapy. However, the enzyme's native hexameric quaternary structure presents challenges for crystallographic studies. The primary objective of this study was to engineer dimeric forms of human GGPPS to facilitate high-resolution crystallographic analysis of its ligand binding interactions. Through site-directed mutagenesis, we disrupted the inter-dimer interactions required for hexamer assembly, generating three stable double-site mutants: Y246D/C247L, Y246D/C205A, and Y246K/C247L. Enzyme assays confirmed that all mutants retained wild-type catalytic activity under both saturating and subsaturating substrate conditions. Differential scanning fluorimetry showed that the mutant proteins had a ~10°C lower melting temperature than the wild-type enzyme but exhibited similar shifts in melting temperature in the presence of the known inhibitors risedronate and zoledronate. Crystallographic analysis of the Y246D/C247L mutant yielded a 2.1 Å resolution structure, providing detailed insights into the binding of isopentenyl pyrophosphate. Closer inspection also revealed the unexpected formation of intermolecular disulfide bonds connecting neighboring dimers, which may explain the enhanced crystallizability of the Y246D/C247L mutant compared to the wild-type and other mutants. These findings highlight the potential of the dimeric mutants as substitutes for wild-type GGPPS in future studies. Optimized dimeric mutants could serve as valuable molecular tools to further our understanding of the enzyme's structural and functional properties and aid in the rational design of novel therapeutic agents targeting GGPPS.

## Introduction

Geranylgeranyl pyrophosphate synthase (GGPPS) catalyzes the production of geranylgeranyl pyrophosphate (GGPP), an isoprenoid intermediate required for protein geranylgeranylation [1]. This post-translational modification attaches the geranylgeranyl group from GGPP to the C-terminal cysteine residue of target proteins [2], which include members of the Ras small GTPase superfamily, such as Rho and Rac proteins [3]. By facilitating the localization of these proteins to cellular membranes, geranylgeranylation is critical for signal transduction pathways that regulate cytoskeletal organization, cell cycle progression, and cell migration [4].

relevant data are within the paper and its Supporting Information files.

**Funding:** This work was supported by a Discovery Grant (RGPIN-2020-00007) from the National Sciences and Engineering Research Council of Canada (https://www.nserc-crsng.gc.ca/) awarded to JP. The sponsor did not have any role in the study design, data collection and analysis, decision to publish, or preparation of the manuscript.

**Competing interests:** The authors have declared that no competing interests exist.

Notably, dysregulation of Rho and Rac small GTPases has been implicated in various stages of cancer development and progression [5]. As a result, inhibiting GGPPS to downregulate protein geranylgeranylation has emerged as a promising strategy for cancer therapy [6].

Recent drug discovery efforts have identified thienopyrimidine-based bisphosphonates as a novel class of GGPPS inhibitors [7, 8]. These compounds have shown effectiveness in blocking protein prenylation in multiple myeloma cells, leading to apoptosis [7]. In a mouse model, administration of a representative inhibitor resulted in the downregulation of small GTPase geranylgeranylation and a reduction in serum levels of monoclonal immunoglobulins, a biomarker of multiple myeloma disease burden [7]. Additionally, the most potent inhibitors in this series demonstrated no significant toxicity in mice and normal human bone marrow cells compared to untreated controls [8]. Characterized by a thienopyrimidine-derived scaffold with an extended side chain, these inhibitors are more lipophilic than current bisphosphonate drugs, such as risedronate and zoledronate. Notably, the large side chain enhances both tissue targeting and cellular uptake and is thought to promote a tight fit within the GGPPS active site [7]. Optimizing enzyme-inhibitor interactions is critical for improving drug candidate potency, but the precise details of thienopyrimidine bisphosphonate binding remain to be elucidated.

An obstacle to characterizing GGPPS-inhibitor interactions is the enzyme's poor propensity to yield diffraction-quality crystals. Currently, only two crystal structures of wild-type (WT) human GGPPS are available, one resolved at 2.7 Å in complex with GGPP, the product of the enzyme, and the other at 2.2 Å in complex with ibandronate, a clinical bisphosphonate drug targeting the GGPPS paralogue, farnesyl pyrophosphate synthase [9, 10]. While these structures have provided valuable insights into the general architecture of the enzyme and its feedback inhibition mechanism, they do not capture interactions with more recently discovered nanomolar inhibitors, such as thienopyrimidine bisphosphonates. This challenge may stem from the unique quaternary structure of human GGPPS. Unlike its homologues that form homodimers, dimers of human GGPPS further associate into a homohexamer, creating an overall structure with a "three-blade propeller" configuration [9]. In contrast, GGPPS predominantly exists as a homodimer in most non-mammalian species, and structures of dimeric homologues have been extensively characterized [11–18]. Given this difference, we hypothesized that a dimeric form of human GGPPS may be more amenable to crystallization, as it could represent a simpler and potentially more uniform structure than the hexameric WT assembly. With only two active sites as opposed to six, dimers are expected to exhibit reduced conformational diversity arising from inhibitor binding, which may induce local structural changes at each binding site or even trigger a global conformational transition of each monomer.

To test our hypothesis, we introduced a site-directed mutation at the dimer-dimer interface of human GGPPS. Key residues involved in inter-dimer interactions were identified (Fig 1A), and a single amino acid substitution replacing Tyr246 with Asp successfully disrupted hexamer formation, thereby creating a mutant protein that exists as a dimer in solution [7]. The potential of this approach was evident, as we routinely obtained crystals of the Y246D mutant protein. However, these crystals exhibited suboptimal diffraction during X-ray analysis, resulting in the highest resolution structure determined at 2.8 Å (PDB ID: 6C56; Fig 1B). This structure revealed an interesting consequence of the hexamer-preventing mutation: increased flexibility of residues around the mutation site led to the formation of a disulfide bond in both subunits of the dimer (Fig 1B). Importantly, the disulfide bonds introduce structural randomness in the $\alpha_1-\alpha_3$ insertion region (Fig 1B), likely contributing to the low resolution of the diffraction data.

In this study, we engineer double-site mutants of human GGPPS to eliminate the Cys247–Cys205 disulfide bond observed in the Y246D mutant. Enzyme assays confirm that all three

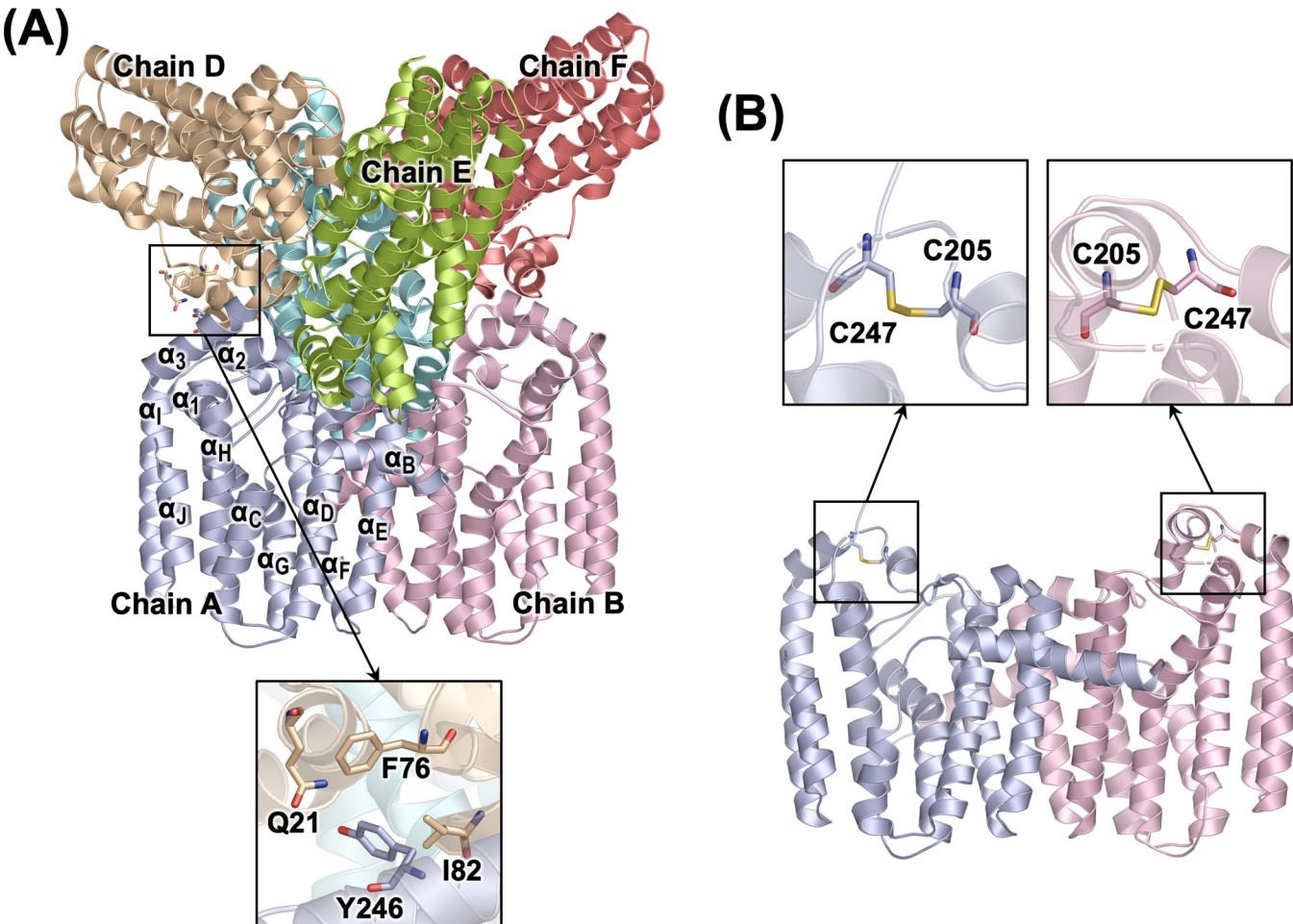

**Fig 1. Crystal structures of human GGPPS.** (A) Homoexameric assembly of WT GGPPS (PDB entry 2Q80). Individual monomeric subunits are depicted in different colors, with a representative monomer labeled to indicate its secondary structure elements. The inter-dimer interface is highlighted by a black box, with a zoomed-in view showing key residues involved in the inter-dimer interactions. (B) Dimeric GGPPS mutant Y246D (PDB entry 6C56). The disulfide bond between C247 and C205 in each subunit is indicated by black boxes, with a close-up view illustrating the disulfide bond formation.

mutants—Y246D/C247L (DL), Y246D/C205A (DA), and Y246K/C247L (KL)—retain WT-level catalytic activity. Differential scanning fluorimetry (DSF) demonstrates that inhibitor binding to these mutants is analogous to that for the WT enzyme, underscoring their suitability for screening studies. Notably, the DL mutant exhibits greater thermal stability than the other two mutants, a characteristic often associated with enhanced crystallizability. Indeed, the DL mutant consistently yields diffraction-quality crystals, enabling us to determine its structure at a resolution of 2.1 Å. This represents the highest resolution structure of human GGPPS to date and provides the first glimpse of the enzyme's interactions with its substrate, isopentenyl pyrophosphate (IPP). We also discuss an unexpected crystallographic artefact and its implications for interpreting the structural data.

## Materials and methods

### Site-directed mutagenesis

The plasmid encoding the human GGPPS Y246D mutant [7] was used as the template for site-directed mutagenesis. The sequences of the primers used for each mutation are as follows:

DL forward - `ATAGATATAAAAAAAGACCTTGTACATTATCTCGAGGAT`
DL reverse - `ATCCTCGAGATAATGTACAAGGTCTTTTTTTTATATCTAT`
DA forward - `AGTGAAAACAAAAGCTTTGCTGAAGATCTGACAGAGGGA`
DA reverse - `TCCCTCTGTCAGATCTTCAGCAAAGCTTTTGTTTTCACT`
KL forward - `GCGCCAGAGAACAGAAAACATAGATATAAAAAAAAAACTTGTACATTATCT`
`CGAGGATGTAGG`
KL reverse - `CCTACATCCTCGAGATAATGTACAAGTTTTTTTTTTATATCTATGTTTTCT`
`GTTCTCTGGCGC`

Mutations were introduced via PCR using a 50 μL reaction containing 25 ng template DNA and 1 μM primers. Following amplification, 3 μL of DpnI-treated PCR product was transformed into 50 μL of *E. coli* DH10B cells (Thermo Fisher Scientific), which were subsequently plated onto 2YT agar plates containing 50 μg/mL kanamycin. Single colonies were selected for miniprep, and the introduction of each mutation was verified by DNA sequencing.

## Expression and purification of WT and mutant GGPPS

The WT and mutant GGPPS proteins were expressed in *E. coli* BL21(DE3) cells (New England Biolabs). The corresponding plasmids were transformed into *E. coli* cells, which were then grown overnight at 37˚C on LB agar plates containing 50 μg/mL kanamycin. Pre-cultures were initiated by inoculating 50 mL of LB containing 50 μg/mL kanamycin with cells from single colonies formed on the agar plates. After overnight incubation at 37˚C, expression cultures were initiated by inoculating 1 L of LB containing 50 μg/mL kanamycin with 10 mL of the pre-culture. The cells were grown at 37˚C until the $OD_{600}$ of the cultures reached 0.6–0.7. Protein expression was induced by adding 1 mL of 1 M isopropyl β-D-1-thiogalactopyranoside and incubating the cultures overnight at 18˚C. The cells were harvested by centrifugation, and the resulting pellets were stored at −20˚C until further use.

For protein purification, the frozen cell pellets were resuspended in a buffer containing 50 mM HEPES (pH 7.5), 500 mM NaCl, 5 mM imidazole, 2 mM β-mercaptoethanol, and 5% glycerol, supplemented with an EDTA-free protease inhibitor cocktail tablet (cOmplete ULTRA, Roche). The cells were lysed by sonication, and the lysate was clarified by centrifugation. The supernatant was then filtered through a 0.45 μm syringe-driven filter unit and applied to a metal ion affinity chromatography column (HisTrap HP 5 mL, Cytiva). GGPPS was eluted using a one-step gradient elution protocol, where the column was initially washed with 5% elution buffer (50 mM HEPES (pH 7.5), 500 mM NaCl, 250 mM imidazole, 2 mM β-mercaptoethanol, and 5% glycerol) until no further protein elution was observed, followed by a 5-column volume elution step with 100% elution buffer. GGPPS-containing fractions were pooled and further purified by size-exclusion chromatography using a Superdex 200 column (HiLoad 16/600 Superdex 200 prep grade, Cytiva) and running buffer consisting of 10 mM HEPES (pH 7.5), 500 mM NaCl, 2 mM β-mercaptoethanol, and 5% glycerol. For crystallization studies, the purified WT and mutant proteins were concentrated to 20 mg/mL and used immediately. For other studies, they were concentrated to 1 mg/mL, flash-frozen in liquid nitrogen, and stored as 100 μL aliquots at −80˚C.

## Enzyme activity assay

The enzymatic activity of the WT and mutant GGPPS proteins was measured using a phosphate release assay [19]. Reactions consisted of 50 mM Tris (pH 7.7), 2 mM $MgCl_2$, 4 mM TCEP, 0.1% Triton X-100, 0.1% BSA, and either 2.1 μM farnesyl pyrophosphate (FPP) and 1.5 μM isopentenyl pyrophosphate (IPP) or 42 μM FPP and 30 μM IPP, at a fixed protein concentration of 2 μg. All reactions were carried out in triplicate at 30˚C in a 96-well flat-bottom

plate, with a total reaction volume of 100 μL per well. Each reaction was initiated by adding 10 μL of FPP and IPP mix to 90 μL of reaction buffer containing all the other assay components after preincubation at 30°C for 5 minutes. The reactions were terminated by quenching with 25 μL of the malachite green working reagent (Malachite Green Phosphate Assay Kit, Sigma) and allowed to develop color for 15–30 minutes before measuring absorbance at 620 nm using a BioTek PowerWave HT Microplate Reader (Agilent).

## Thermal shift assay

DSF was used to assess the thermal stability of the WT and mutant GGPPS proteins. Experiments were carried out in triplicate in a white 96-well qPCR plate, with a total reaction volume of 25 μL per well. Each reaction contained 10 mM HEPES (pH 7.5), 100 mM NaCl, 3 mM $MgCl_2$, 0.5 mM TCEP, 5 μM protein, and 5× SYPRO Orange dye (Thermo Fisher Scientific). When present, the concentration of the added inhibitor ranged from 0.01 μM to 2000 μM. Melting curves were generated by increasing the temperature stepwise at a rate of 1°C/min from 25°C to 95°C in a CFX Connect Real-Time PCR System (Bio-Rad). Melting temperatures ($T_m$) were determined by Gaussian peak-fitting the first negative derivative of each melting curve in GraphPad Prism (GraphPad Software). To determine the dissociation constant ($K_d$) for the protein-inhibitor binding, the change in $T_m$ was plotted against inhibitor concentration and fit to the one-site saturation binding model implemented in GraphPad Prism.

## Crystallization, X-ray diffraction, and structure determination

A crystal of the DL mutant was grown at 22°C by vapour diffusion in a sitting drop composed of 1 μL of protein solution (10 mM HEPES (pH 7.5), 500 mM NaCl, 2 mM β-mercaptoethanol, 5% glycerol, 3 mM $MgCl_2$, 2 mM zoledronate, 2 mM IPP, and 15 mg/mL GGPPS DL) and 1 μL of crystallization buffer (2.2 M $(NH_4)_2SO_4$ and 200 mM $(NH_4)_2HPO_4$). The protein solution was prepared by mixing the purified enzyme stock with ligands ($MgCl_2$, zoledronate, and IPP) and incubated on ice for 30 minutes prior to setting up the crystallization drops. Diffraction data were collected from this crystal at 100 K using synchrotron radiation (Beamline 08ID-1, Canadian Light Source). The data set was processed with *XDS* [20], *POINTLESS* [21], and *CCP*4 [22] via the *xia*2 autoprocessing interface [23]. Initial phase information was obtained by molecular replacement using *Phaser* [24], with a ligand/solvent-omitted search model derived from PDB entry 2Q80. After iterative cycles of manual model building using *Coot* [25] and automated refinement by *REFMAC*5 [26], the final structure model was deposited in the PDB under the entry code 9CSL. A structure validation report provided by the PDB is included in the supporting information (S1 File). Data collection and structure refinement statistics are summarized in Table 1.

# Results and discussion

## Oligomeric states of GGPPS mutants

Oligomeric states of the WT and mutant GGPPS proteins were assessed during the size exclusion chromatography step of the purification process. The chromatogram for the WT protein shows a sharp peak at an elution volume of 60.6 mL (Fig 2A), corresponding to a molecular mass of 219,188 Da (inset, Fig 2A), consistent with the theoretical value of 224,544 Da expected for a homohexameric complex composed of 37,424-Da subunits. For the DL mutant, the chromatogram shows an elution volume of 72.0 mL (Fig 2A), corresponding to a molecular mass of 80,564 Da (inset, Fig 2A). This value aligns with the dimeric mass of 74,848 Da, confirming the successful disruption of hexamer formation in this mutant. In contrast, the DA and KL mutants exhibited unexpected elution volumes of 68.7 mL and 66.8 mL, respectively (Fig 2A),

**Table 1. Data collection and structure refinement statistics.**

| Data collection[a] | |
|---|---|
| Wavelength (Å) | 0.97949 |
| Space group | $P622$ |
| Unit cell dimension (Å) | $a = b = 191.11, c = 96.63$ |
| Unit cell angles (°) | $\alpha = \beta = 90, \gamma = 120$ |
| Resolution range (Å) | 95.56–2.10 (2.14–2.10) |
| Completeness (%) | 99.9 (99.1) |
| Redundancy | 19.6 (16.9) |
| $I/\sigma(I)$ | 20.4 (1.6) |
| $R_{pim}$ | 0.023 (0.419) |
| $CC_{1/2}$ | 1 (0.665) |
| **Structure refinement** | |
| No. of reflections | 60724 |
| $R_{work}/R_{free}$ | 0.179/0.216 |
| No. of non-H atoms in model | |
| Protein | 4599 |
| Ligand | 75 |
| Water | 424 |
| Total | 5098 |
| Overall $B$-factor (Å$^2$) | 46.68 |
| R.m.s. deviations | |
| Bonds (Å) | 0.009 |
| Angles (°) | 1.725 |
| Ramachandran plot | |
| Most favoured (%) | 99 |
| Allowed (%) | 1 |
| Outlier (%) | 0 |

[a]Values in parentheses are for the highest resolution shell.

with corresponding molecular masses of 107,017 Da and 127,012 Da (inset, Fig 2A). These values do not match any predicted oligomeric state of GGPPS, as they are larger than the dimeric mass of 74,848 Da but smaller than the tetrameric mass of 149,696 Da. Notably, the elution peaks for these mutants are significantly broader, with widths of ~20 mL, compared to ~10 mL observed for the WT and DL proteins (Fig 2A). Subsequent SDS-PAGE analysis confirmed that the WT and mutant proteins were all expressed with monomeric molecular masses consistent with the theoretical value of ~37 kDa and that the purified proteins were of high purity, free from significant degradation products or contaminating proteins (Fig 1B).

Taken together, these results underscore the importance of specific interactions at the dimer-dimer interface in stabilizing the hexameric assembly of hGGPPS, highlighting the effectiveness of the Y246D/C247L double mutation in producing a stable dimeric form of the enzyme. In contrast, the unexpected elution profiles observed for the DA and KL mutants remained unclear at this stage of our study. It was hypothesized that the distinct patterns might have resulted from non-specific interactions introduced by the Y246K and C205A mutations, potentially leading to protein aggregation. Alternatively, altered hydrodynamic behaviour caused by increased conformational flexibility resulting from these mutations could have contributed to the observed differences. These possibilities were further explored in subsequent experiments.

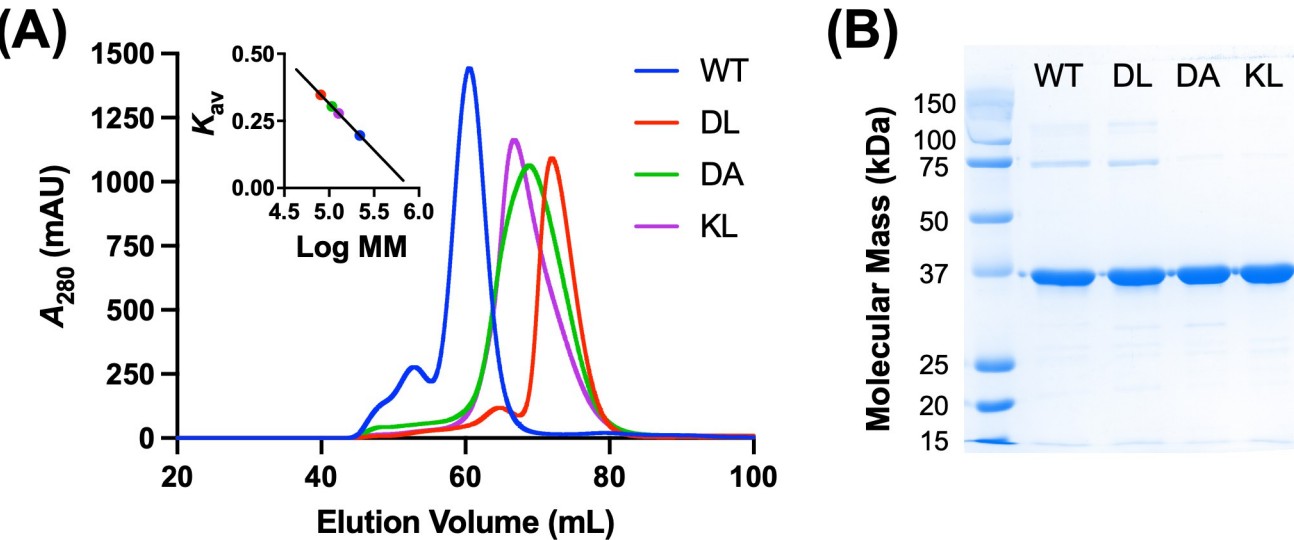

**Fig 2. Purification of WT and mutant GGPPS proteins.** (A) Size exclusion chromatograms of WT and mutant GGPPS proteins, with each chromatogram colour-coded as indicated in the legend. The inset shows the standard curve used to estimate the molecular masses of the eluted proteins. (B) SDS-PAGE analysis of purified WT and mutant GGPPS samples, representing the central 50% of peak fractions pooled from the size exclusion chromatography runs.

### Catalytic activity of GGPPS mutants

Next, we examined the catalytic functionality of our mutant proteins. GGPPS synthesizes GGPP by condensing the allylic substrate farnesyl pyrophosphate (FPP) with the homoallylic substrate IPP, releasing inorganic pyrophosphate as a by-product. The catalytic activity was measured in a coupled enzyme assay employing inorganic pyrophosphatase by detecting the release of inorganic phosphate via the malachite green dye [19]. The assay was conducted under two substrate conditions: 0.5 and 10 times the literature $K_m$ values [9]. Interestingly, at lower, subsaturating substrate concentrations ([FPP] = 2.1 μM, [IPP] = 1.5 μM), the catalytic activity of all three mutant proteins exceeded that of WT GGPPS. The initial rates for the DL, DA, and KL mutants were 0.203 μM/min, 0.184 μM/min, and 0.218 μM/min, respectively, compared to 0.139 μM/min for the WT control (Fig 3). At higher, saturating substrate concentrations ([FPP] = 42 μM, [IPP] = 30 μM), the DL and KL mutants exhibited catalytic activity similar to that of the WT enzyme, with initial rates of ~0.29 μM/min (Fig 3). The DA mutant showed slightly lower activity in this substrate concentration range, with an initial rate of 0.248 μM/min; however, this difference was not statistically significant (Fig 3).

The observed rate data demonstrate that our mutant proteins retain functional viability across a range of substrate concentrations. The introduced mutations do not impair enzymatic activity, while they clearly affect the oligomeric assembly of the proteins. This observation suggests that the mutant proteins could be useful in functional studies, such as inhibitor screening, especially since the mutations are located on the protein surface, distant from the active site. Consistent with this, previous studies have reported no allosteric or cooperative behaviour between the individual subunits of GGPPS. Moreover, the finding that the mutant proteins exhibit increased catalytic activity under sub-saturating substrate conditions is intriguing. Although the mechanism underlying this effect remains unclear—particularly as the mutations are located > 15 Å from the nearest substrate-binding site—a conformational mechanism may be involved. For example, structural changes required for substrate binding or catalysis may occur more freely in the dimer mutants, whereas in the WT hexamer, such movements could be constrained by inter-dimer interactions.

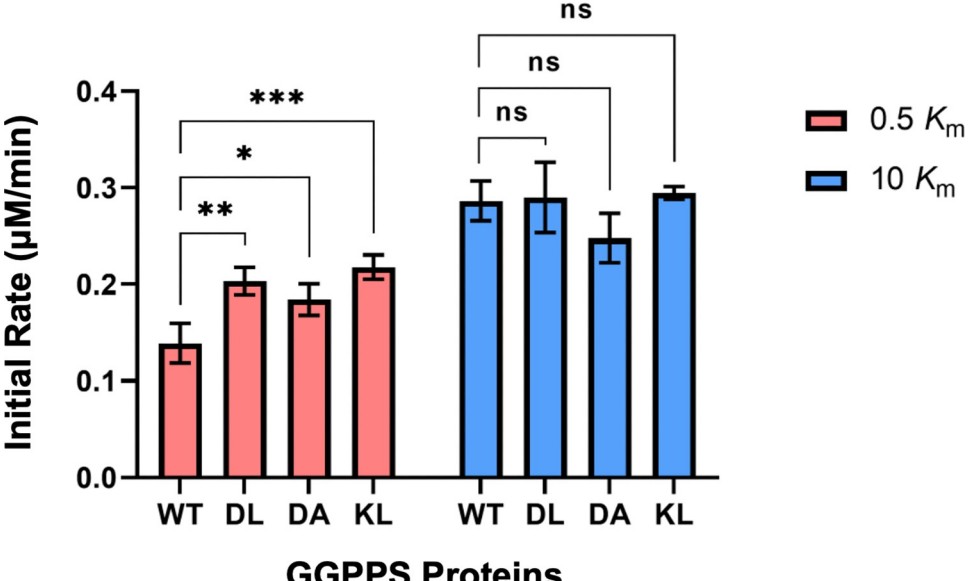

**Fig 3. Catalytic activity of WT and mutant GGPPS proteins.** Initial rates of GGPP synthesis by WT GGPPS and the DL, DA, and KL mutants were measured under two substrate conditions: subsaturating (0.5 $K_m$, red) and saturating (10 $K_m$, blue) concentrations of FPP and IPP. Error bars represent standard deviation ($n = 3$). $P$ values were determined by one-way ANOVA and are indicated by asterisks: $^*p = 0.0227$, $^{**}p = 0.0033$, and $^{***}p = 0.0009$.

## Thermal stability of GGPPS mutants

Subsequently, we measured the $T_m$ of our GGPPS mutants, both in the absence and presence of known inhibitors, to gain insights into the structural stability of these proteins and their interactions with inhibitors. DSF results showed that the DL, KL, and DA mutants have $T_m$ values of 50.2˚C, 49.3˚C, and 49.1˚C, respectively (Fig 4; melting curves shown in S1 Fig). In comparison, WT GGPPS exhibited a $T_m$ of 59.6˚C, ~10˚C higher than those of the mutant proteins (Fig 4; S1 Fig). The lower $T_m$ values observed for the mutants were expected and reflect their reduced oligomeric states compared to the hexameric WT protein. These results also indicate that all three mutant proteins exist as dimers in solution. Previous size exclusion results clearly demonstrated the DL mutant to be dimeric, while the estimated molecular masses for the KL and DA mutants fell between those expected for a dimer and a tetramer. However, if the latter two mutants formed larger complexes, they would exhibit higher melting temperatures. Instead, their $T_m$ values are ~1˚C lower than that of the DL mutant, effectively ruling out the possibility of higher-order oligomers and aggregates, a conclusion further supported by native-PAGE analysis (S2 Fig). The decrease in $T_m$ values suggests slightly less stable structures for these mutants, consistent with increased conformational flexibility, as reflected in their size exclusion profiles.

In the presence of the inhibitors risedronate and zoledronate, the $T_m$ of both WT and mutant proteins increased in a dose-dependent manner (Fig 5; S3 Fig). In thermal shift assays, an increase in $T_m$ reflects inhibitor binding, which stabilizes the structure of the target protein, making it more resistant to thermal denaturation. Analysis of the $T_m$ change ($\Delta T_m$) as a function of inhibitor concentration yielded $K_d$ values of 2.53 μM and 1.64 μM for risedronate and zoledronate, respectively, for WT GGPPS (Fig 5; Table 2), which align well with the previously reported values of 3.35 μM for risedronate and 2.76 μM for zoledronate [10]. In comparison, the $K_d$ values for the dimer mutants are 2–3 times higher (Fig 5; Table 2), indicating a corresponding decrease in binding affinity between the mutants and inhibitors. Nevertheless, a

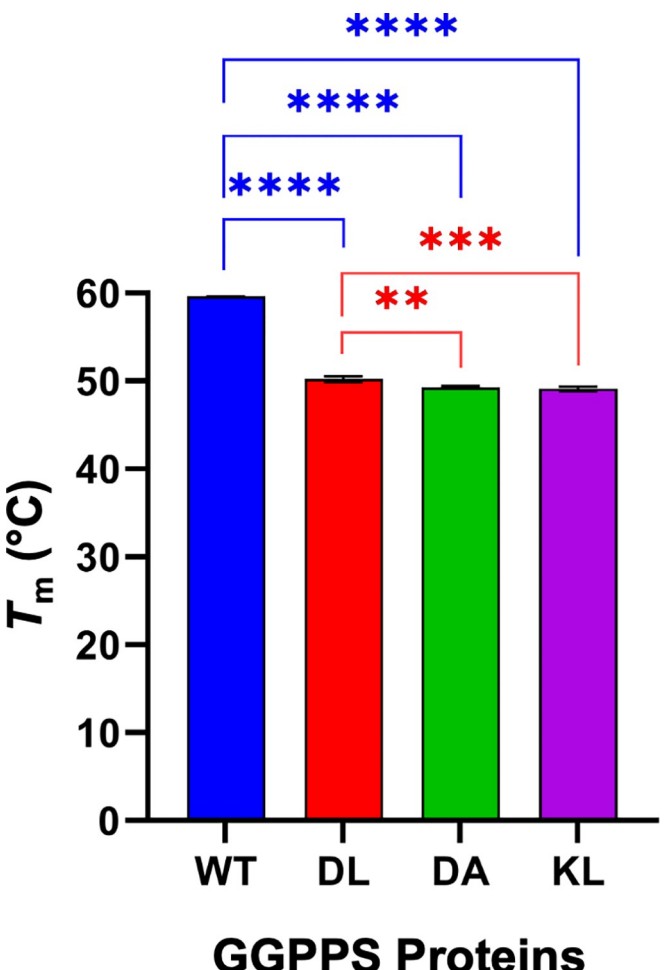

**Fig 4. $T_m$ of WT and dimer mutant GGPPS determined by DSF.** Error bars represent standard deviation ($n = 3$). $P$ values were determined by one-way ANOVA and are indicated by asterisks: **$p = 0.0033$, ***$p = 0.009$, and ****$p < 0.0001$.

consistent trend was observed across both WT and mutant proteins, with $K_d$ values slightly higher for zoledronate than for risedronate. These findings suggest that the mutant proteins retain a functional active site capable of inhibitor binding, albeit with slightly lower affinities compared to WT GGPPS. The results also highlight the potential utility of dimer mutants in DSF-based screening studies. The lower basal $T_m$ of the mutants may enhance assay sensitivity, particularly in initial screening campaigns where compounds are tested at high concentrations. For example, the $T_m$ changes induced by high concentrations of risedronate and zoledronate are 2–3˚C greater for the DA and KL mutants than for the WT protein (Fig 5). However, while the dimer mutants may be useful for identifying initial hits, further characterization of promising compounds, including $K_d$ determination and mechanistic studies, should be conducted using the WT protein to ensure the most accurate assessment of inhibitor potency and selectivity.

## Crystal structure of mutant GGPPS in complex with IPP

Next, we attempted to crystallize the GGPPS dimer mutants to determine their atomic structures by X-ray crystallography. Crystals of the DL mutant were successfully obtained, with the

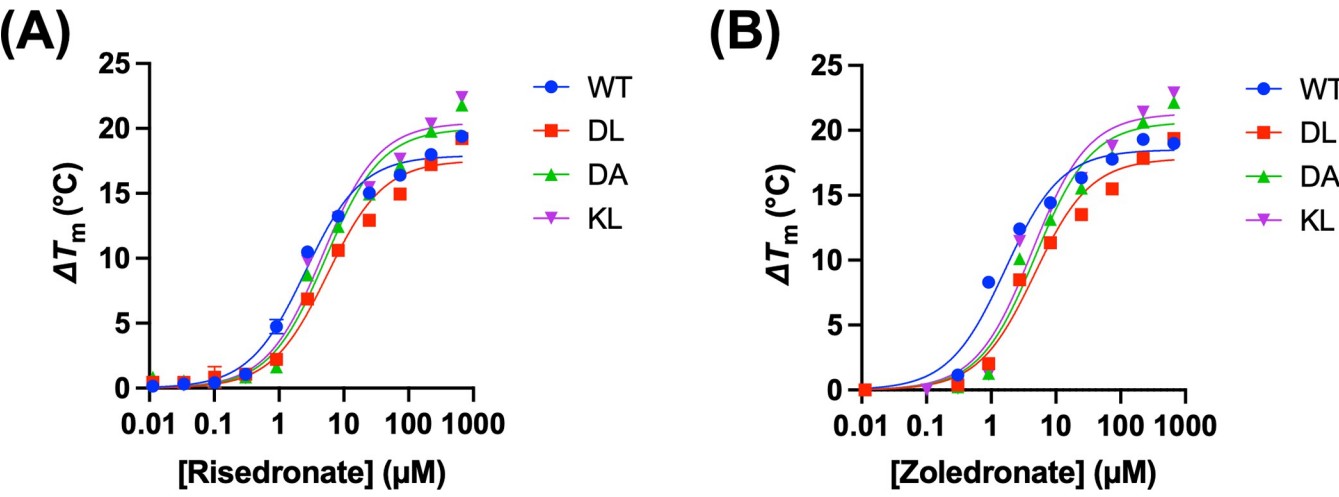

**Fig 5. Thermal stability of WT and dimer mutant GGPPS in the presence of known inhibitors.** (A) Changes in $T_m$ measured by DSF as a function of increasing risedronate concentration. (B) Changes in $T_m$ measured by DSF as a function of increasing zoledronate concentration. The binding isotherms for each protein are colour-coded according to the legend. Data points represent mean ± standard deviation ($n = 3$).

highest quality crystal diffracting to a resolution of 2.1 Å. The crystals exhibited a hexagonal plate morphology (S4 Fig), with typical dimensions of approximately 150 × 150 × 30 μm. Crystals appeared within 2–3 weeks and remained stable for at least 6–8 weeks without noticeable deterioration. These crystals exhibited symmetry corresponding to space group *P*622, unlike the *I*422 space group observed for the parent Y246D mutant (PDB ID: 6C56). The distinct symmetry groups and lattice parameters reflect significant differences in crystallographic packing, despite only a single amino acid difference. As with Y246D, the asymmetric unit of the DL crystal contains a single homodimer, consistent with its oligomeric state confirmed by earlier solution studies. However, despite extensive crystallization trials, the DA and KL mutants did not yield crystals of sufficient quality for diffraction experiments. This difficulty highlights the substantial impact a single amino acid can have on protein structural properties and may be attributed to the reduced stability of these mutants, as suggested by their hydrodynamic profiles and $T_m$ values.

Interestingly, the DL crystal did not contain a bound inhibitor molecule, despite the presence of zoledronate in the crystallization sample. However, electron density observed in a difference map ($F_o$–$F_c$, contoured at the 3σ level) clearly indicated the presence of IPP within its anticipated binding site in one subunit of the dimer mutant (chain A; PDB ID: 9CSL). IPP binding is primarily stabilized by interactions with basic residues in the binding site (Fig 6A). The guanidino group of Arg73 and the backbone amide nitrogen of Gln26 engage in water-mediated hydrogen bonds with the β-phosphate of IPP. Direct interactions include salt bridges between Arg74 and the β-phosphate, Arg28 and the α-phosphate, and Lys 25 with both α- and β-phosphates. Additionally, the backbone amide nitrogen of Lys25 and the side chain epsilon

**Table 2. Inhibitor binding affinities of GGPPS variants determined by DSF.**

| Protein | $K_d$ Risedronate (μM) | $K_d$ Zoledronate (μM) |
|---------|------------------------|------------------------|
| WT | 2.53 ± 0.211 | 1.64 ± 0.214 |
| DL | 5.59 ± 0.564 | 4.67 ± 0.515 |
| DA | 5.24 ± 0.578 | 4.89 ± 0.605 |
| KL | 4.70 ± 0.553 | 4.27 ± 0.588 |

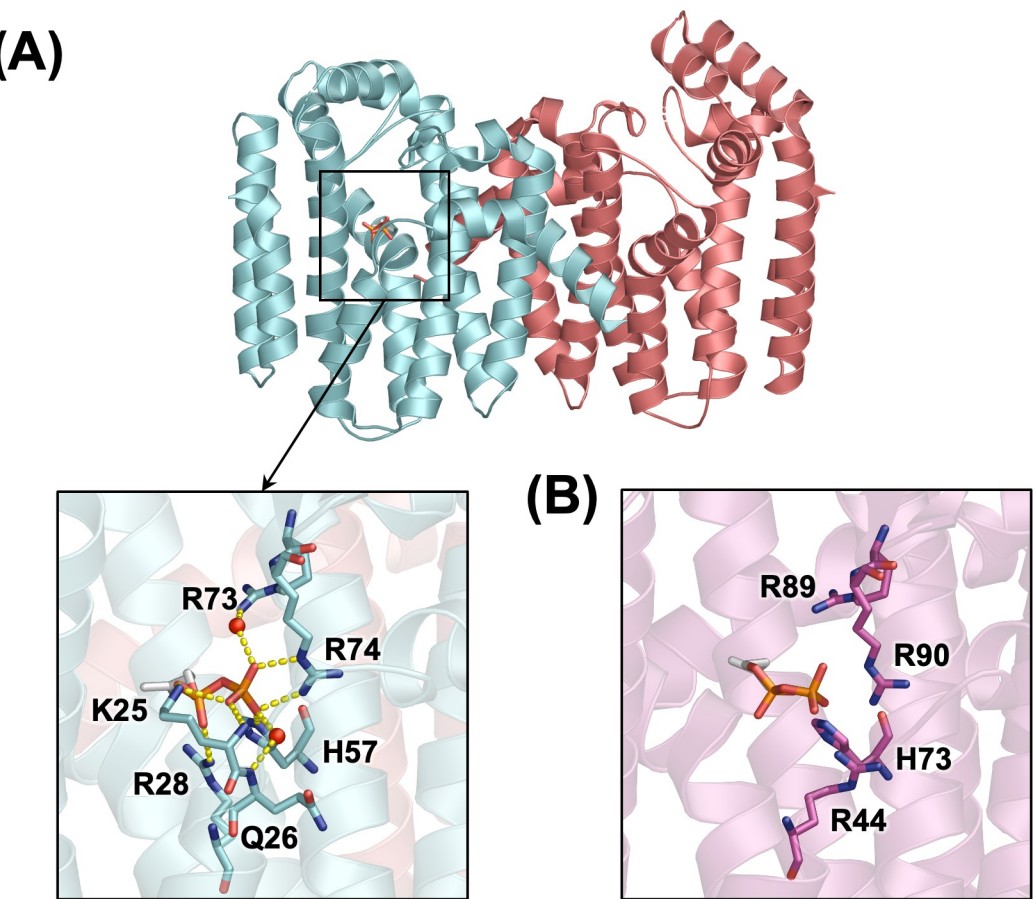

**Fig 6. Crystal structures of GGPPS dimers.** (A) Structure of the human GGPPS DL mutant (PDB ID: 9CSL), with each subunit of the dimer highlighted in different colours (chain A in cyan and chain B in salmon). The inset provides a close-up view of the IPP binding site, showing key residues involved in the binding interactions. (B) IPP binding site in *S. cerevisiae* GGPPS (chain A; PDB ID: 2E8T) [11]. The conserved basic residues that contribute to IPP binding are represented.

nitrogen of His57 form hydrogen bonds with β-phosphate oxygen atoms, further stabilizing the binding interaction. Finally, the isoprenyl tail of IPP makes van der Waals contacts with the side chains of Gln185 and Thr152.

While the current structure is the first to demonstrate IPP binding in human GGPPS, similar binding interactions have been observed in orthologues from other species, such as *S. cerevisiae*. Comparison with the *S. cerevisiae* GGPPS structure (PDB ID: 2E8T) reveals that many key residues involved in IPP binding are conserved in both identity and conformation (Fig 6B). A notable difference is in the conformation of Arg28 in the human enzyme, which binds the α-phosphate of IPP, whereas in *S. cerevisiae*, the corresponding residue (Arg44) binds the β-phosphate (Fig 6B). Additionally, Lys41, the equivalent of Lys25 in the human enzyme, is part of a five-amino acid disordered region in the yeast homologue.

Notably, the new structure confirms the resolution of the issue we originally aimed to address. In our previous study, the Y246D dimer mutant formed intramolecular disulfide bonds between Cys247 and Cys205 in both subunits, which we suspected were interfering with crystallographic packing. In the DL mutant, where Cys247 was replaced by Leu, these disulfide bonds were eliminated. However, this substitution introduced an unexpected crystallographic artefact: the formation of intermolecular disulfide bonds between the Cys205 residues of the

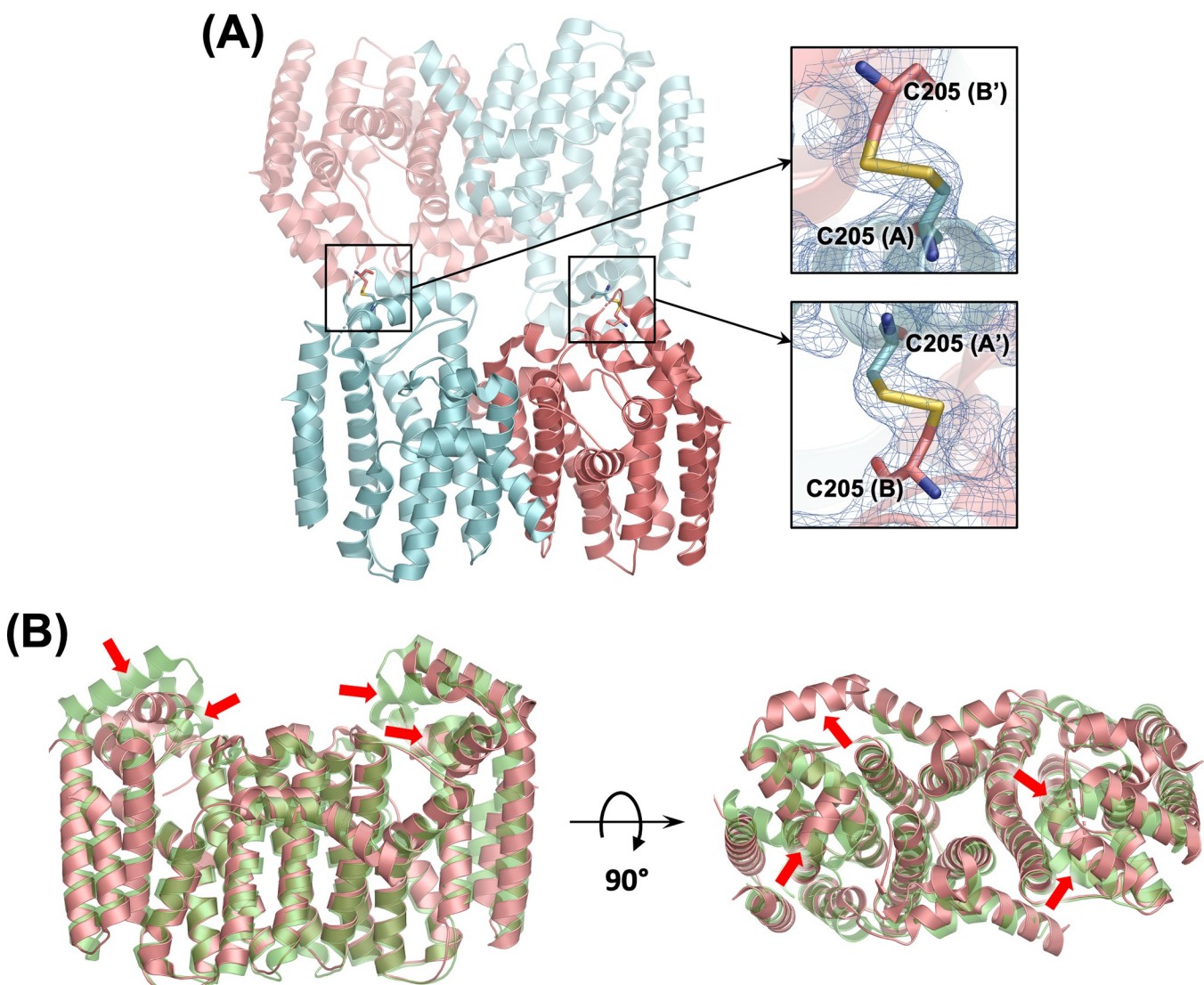

**Fig 7. Disulfide bond formation in the GGPPS DL mutant.** (A) Covalent linkage between symmetry-related dimer units. The upper dimer is displayed semi-transparently, with monomeric subunits shown in cyan (chain A) and salmon (chain B). Insets provide zoomed-in views of the intermolecular disulfide bonds. The blue mesh represents the $2F_o - F_c$ electron density map contoured at 1 σ. (B) Superimposition of the DL mutant (salmon) with the WT GGPPS dimer (semi-transparent green, chains A and B extracted from the hexamer structure; PDB ID: 2Q80). Red arrows indicate regions of conformational variation in the front view (left) and top view (right).

asymmetric unit and those of a symmetry-related dimer (Fig 7A). This suggests that the inclusion of 2 mM β-mercaptoethanol was insufficient to prevent cysteine oxidation during the 2–3 week crystallization period, particularly under the high protein concentration used. The resulting disulfide bonds appear to have a dual effect on the crystal structure. First, they may facilitate nucleation by bringing the dimer units together during crystallization, as evidenced by the DL mutant's higher propensity for crystallization compared to the WT and other dimer mutants. Second, the disulfide bonds disrupt the conformation of the $\alpha_1$–$\alpha_3$ insertion region (Fig 7B), which acts as both an inter-dimer interface and as a partial lid for the enzyme's active site. This structural change was unexpected, as our AlphaFold [27] prediction for the DL mutant resulted in a structure nearly identical to that of WT GGPPS (S5 Fig).

The intermolecular disulfide bonds observed in the DL mutant structure may have significant implications for understanding inhibitor binding. For example, the binding of bisphosphonate inhibitors to the closely related paralogue farnesyl pyrophosphate synthase induces a large conformational change that brings the enzyme's two lobes together, resulting in a tightly closed conformation that enables high-affinity binding [28–31]. Although a similar induced-fit mechanism in GGPPS remains unconfirmed due to the lack of relevant structures, it is conceivable that if such a mechanism exists, the observed intermolecular disulfide bonds could inhibit the necessary conformational change, preventing tighter binding of bisphosphonate compounds—potentially explaining the absence of bound zoledronate in the current structure. Interestingly, a region of unexplained electron density was observed within the active site of the DL mutant, overlapping with the space typically occupied by bound bisphosphonates (S6 Fig). Moreover, an alternate conformation of Asp64 required for bisphosphonate binding was identified in this structure (S6 Fig). Therefore, while the observed density is not sufficiently defined to confirm the identity of a potentially bound ligand, it may suggest partial occupancy of zoledronate in the structure.

## Conclusion

GGPPS plays a crucial role in protein prenylation and is implicated in various diseases, making it an attractive target for therapeutic intervention. Given the challenges associated with the structural characterization of this enzyme, protein engineering offers promising strategies to facilitate the rational design and optimization of inhibitors. For example, a dimeric form of GGPPS may provide a simpler system for investigating enzyme-inhibitor interactions. In this study, we generated catalytically active dimeric mutants of GGPPS, which could serve as surrogates for the WT enzyme in drug discovery research. One of these mutants yielded a 2.1 Å resolution crystal structure, leading to the first characterization of IPP binding in human GGPPS. This structure also revealed an unexpected consequence of the engineered mutation—the formation of disulfide bonds between dimer units in the crystal. While increased use of reducing agents during crystallization or further mutagenesis could address this issue, insights from the current mutants highlight the profound impact that even a single amino acid mutation can have on enzyme structure, stability, and dynamics. Ultimately, optimized dimeric forms of GGPPS could serve as valuable molecular tools in therapeutic development, providing a more reliable structural platform for screening and designing novel inhibitors.

## Supporting information

**S1 File. PDB structure validation report for entry 9CSL.**
(PDF)

**S1 Fig. DSF melting curves for determining the $T_m$ of GGPPS proteins.**
(TIF)

**S2 Fig. Native-PAGE analysis of dimeric GGPPS mutants.**
(TIF)

**S3 Fig. DSF melting curves of GGPPS proteins in the presence of risedronate and zoledronate.**
(TIF)

**S4 Fig. Crystals of the GGPPS DL mutant.**
(TIF)

**S5 Fig. AlphaFold-predicted structure of the GGPPS DL mutant.** The AlphaFold-predicted DL mutant structure (purple) is superimposed onto the WT GGPPS structure (transparent green, PDB ID: 2Q80). The bottom image provides a top view of the same superposition. (TIF)

**S6 Fig. Close-up view of the active site structure of the GGPPS DL mutant.** The DL mutant structure (magenta) is overlaid onto the ibandronate-bound WT GGPPS structure (semi-transparent cyan, ibandronate in orange; PDB ID: 6R4V). The green mesh represents the $F_o - F_c$ difference map, contoured at 3 σ. Green spheres indicate magnesium ions co-bound with the bisphosphonate. The unidentified density overlaps with the position of the bisphosphonate side chain. Two alternate conformations of Asp64 were identified in the DL mutant structure. (TIF)

**S1 Raw images. Original images corresponding to all gel results presented in this article.** (A) Original gel image for Fig 2B. Lanes loaded with WT and mutant protein samples are labeled. Lanes overloaded with identical samples and excluded from the final figure are marked with X. (B) Original gel image for S2 Fig. Lanes loaded with mutant protein samples are labeled, and empty lanes are marked with E. Empty lanes excluded from the final figure are marked with X. Both gels were imaged using a BioRad ChemiDoc Imaging System. (PDF)

**S1 Raw data. All values used to build graphs, as well as to determine means, standard deviations, and other measures reported in this study.** (XLSX)

## Acknowledgments

We would like to express our sincerest gratitude to Barry Sleno, former technician in the Berghuis Lab at McGill University, for his valuable technical assistance with the site-directed mutagenesis experiments.

## Author Contributions

**Conceptualization:** Jaeok Park.

**Formal analysis:** Sean J. Ezekiel, Mackenzie Searle, Jaeok Park.

**Funding acquisition:** Jaeok Park.

**Investigation:** Sean J. Ezekiel, Mackenzie Searle, Jaeok Park.

**Supervision:** Jaeok Park.

**Visualization:** Sean J. Ezekiel.

**Writing – original draft:** Sean J. Ezekiel.

**Writing – review & editing:** Jaeok Park.

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
