## [Decision Letter · Decision Letter 0]

21 Oct 2024

PONE-D-24-41093Engineering dimer mutants of human geranylgeranyl pyrophosphate synthasePLOS ONE

Dear Dr. Park,

Thank you for submitting your manuscript to PLOS ONE. We have now received reports from 2 reviewers. Based on their comments and after careful consideration, we feel that it has merit but does not fully meet PLOS ONE’s publication criteria as it currently stands. Therefore, we invite you to submit a revised version of the manuscript that addresses all the points raised by the reviewers (see their reports below). Editorially, our reading of the reviewer reports indicates that further major experiments are likely not required. Instead, including missing controls, minor experiments, caveats, and text discussion will likely satisfy the referees.

We look forward to receiving your revised manuscript.

Kind regards,

Yoni Haitin, Ph.D.

Academic Editor

PLOS ONE

Journal Requirements:

Reviewers' comments:

Reviewer's Responses to Questions

**Comments to the Author**

1. Is the manuscript technically sound, and do the data support the conclusions?

Reviewer #1: Yes

Reviewer #2: Partly

2. Has the statistical analysis been performed appropriately and rigorously? 

Reviewer #1: Yes

Reviewer #2: No

3. Have the authors made all data underlying the findings in their manuscript fully available?

Reviewer #1: Yes

Reviewer #2: Yes

4. Is the manuscript presented in an intelligible fashion and written in standard English?

Reviewer #1: Yes

Reviewer #2: Yes

5. Review Comments to the Author

Reviewer #1: In this manuscript, the authors have engineered dimeric mutants of human geranylgeranyl pyrophosphate synthase (GGPPS) enzyme to overcome the widely reported challenges faced with crystallizing the enzyme in its native hexameric form. They used site-directed mutagenesis to disrupt the native hexameric structure, while retaining catalytic activity, resulting in three stable dimeric mutants, namely Y246D/C247L, Y246D/C205A, and Y246K/C247L. Crystallographic analysis of the Y246D/C247L mutant provided structural insights into ligand binding, suggesting these dimeric forms could aid in future drug design targeting GGPPS. Overall, the authors have succeeded in their goal of establishing a simple methodology for crystallizing the otherwise difficult to crystallize human GGDPS enzyme, and therefore this study has the potential to significantly benefit future GGDPS inhibitor binding studies.

The following comments would improve the presentation:

• In Table 1, the CC1/2 value of the highest shell is low (0.665). Can the authors explain why they expect the highest resolution shell to have such low CC1/2? Or could they re-merge the data to a lower resolution to get a better CC1/2?

• Page 11 last paragraph “nonspecific interactions introduced by the Y246K and C205A mutations, potentially leading to protein aggregation. “ DLS would be an effective measure of the aggregation states of each sample coming off the column.

• Page 12, Catalytic activity of GGPPS mutants, first paragraph. Does not mention the wavelength at which malachite green dye absorbs.

• Page 12, last paragraph, “do not impair enzymatic activity”. The authors should rephrase this as they state under Catalytic activity of GGPPS mutants on page 12, “However, the DA mutant showed somewhat lower activity in this substrate concentration range,”

• Page 15 second paragraph second line “However, electron density data clearly indicated…”. Should include the sigma level used to make this determination.

• Page 18 Conclusion line 4 “For example, a dimeric form of GGPPS may provide a simpler system for investigating enzyme-inhibitor interactions. “ The dimeric form appears to be more easily crystallizable- but are we worried about introducing artefactual disulfide bonds (that typically do not exist in the wt) to mess with inhibitor binding?

• Page 16 Conclusion line 10 “ While increased use of reducing agents during crystallization or further mutagenesis could address this issue,” The authors need to explore this more. Use different reductants (DTT?) to see if they can prevent the extra bonds from forming, so as to maintain an inhibitor binding environment similar to wt.

• Figure 1A. Should mark the chains in the ribbon diagram. Then authors can refer to the same coloring scheme throughout the text.

• Figure 5 the figure labels are unclear. The blue inset (A) shows some hydrogen bonds which are a little hard to see.

• Methods description comments

• Was the His-Tag cleaved?

• Please describe the Imidazole gradient used for elution, starting and ending concentrations and number of column volumes, What is the composition of the elution buffer?

• What is the crystal morphology (a crystal picture would be helpful)? What were the crystal dimensions and how long it took for crystals to appear. Were the crystals stable over time, or did they dissolve over time?

• How long was the protein incubated with ligand for before setting up the trays?

Reviewer #2: Ezekiel et al. report the engineering and characterization of dimeric mutants of human GGPPS for structural and functional analyses. They specifically highlight the potential of these constructs to promote identification and optimization of novel inhibitors. While the experimental work and data analysis are sound, several points should be addressed by the authors:

1. The authors state that “only two crystal structures of wild-type (WT) human GGPPS are available, one resolved at a moderate resolution of 2.7 Å, and neither in complex with a relevant inhibitor molecule”. I assume that the second structure is 6R4V, obtained at 2.2 Å resolution with a bound bisphosphonate (as attempted in this study). Therefore, this statement is unclear and seems to be misleading.

2. The mutants affect the catalytic activity, at least under the limited conditions inspected here. However, it is concluded that “the mutant proteins could be useful in functional studies, such as inhibitor screening, and aligns with the fact that the mutations are located on the protein surface, distant from the active site..”. I do not refute the possibility to use these constructs for inhibitor screening, but it seems that the active site is affected and this may have implications for drug discovery. Do the authors have any thoughts on how the mutations may affect catalytic activity and active site?

3. TSA analysis – the change in Tm values is very much overinterpreted in my opinion. The change in Tm may be consistent with much of the claims, but does not prove most of them. For example, reduction in Tm is not a direct indication for change in the oligomeric state or conformational flexibility. Please reconsider the extent of interpretation in this section.

4. Similar to #2, the Kd for bisphosphonates is also altered to some extent by the mutations in the DSF studies. The entusiasm to use dimers in screening campaigns as suggested by the authors is less clear as the hexameric protein seems to behave well in solution and there are differences in affinity, so good inhibitors may be missed by using the mutant. Consider elaborating the pros and cons of this approach.

5. The major problem is that the constructs were design to obtain structures of GGPPS bound to inhibitors, yet an inhibitor was not clearly identified despite the 2.1 Å resolution. With the 2.2 Å resolution structure of the hexameric enzyme bound to ibandronate in hand, the contribution of the designed dimer is unclear.

6. The lack of identified zolendronate also highlights the potential distortion of the active site by the mutation and resulting crystallographic artefacts (which obviously couldn’t have been predicted by alphafold…) which are deleterious for structural investigations aimed at improving small molecules binding. Did the authors try to fit a zolendroate molecule with partial occupancy?

Minor:

1. Abstract – differential scanning, and not scattering, fluorimetry.

2. Methods – did you mean 420 μM FPP and 300 μM IPP or perhaps 42 and 30, respectively? Otherwise I assume that the authors meant 100 times, and not 10 times, the Km for the saturating conditions.

3. Statistical analysis should be provided for differences in catalytic activity, etc.

4. How did the authors made sure that the measured rates are initial? How was that defined?

6. PLOS authors have the option to publish the peer review history of their article (what does this mean?). If published, this will include your full peer review and any attached files.

Reviewer #1: No

Reviewer #2: No

---

## [Author Response · Author response to Decision Letter 0]

19 Dec 2024

We have thoroughly reviewed the referee’s insightful comments and have revised our manuscript to address their suggestions. In accordance with the submission guidelines, we have provided a version of the manuscript with all changes clearly highlighted. We sincerely thank the reviewers for their constructive feedback, which has significantly enhanced the quality and clarity of our work. Below, we present our detailed point-by-point responses to each of the reviewers’ comments. Page and line references correspond to the tracked-changes version of the revised manuscript.

Reviewer #1

• In Table 1, the CC1/2 value of the highest shell is low (0.665). Can the authors explain why they expect the highest resolution shell to have such low CC1/2? Or could they re-merge the data to a lower resolution to get a better CC1/2?

We respectfully contend that the CC1/2 value of 0.665 for the highest-resolution shell is not low. In fact, this value is well above the cut-off thresholds commonly used. For example, xia2 (the software used to process our data) recommends a cut-off value between 0.3 and 0.5. Current crystallographic practices frequently extend the acceptable range for CC1/2 of the highest resolution shell to 0.1–0.2. Therefore, we believe our chosen resolution cut-off is appropriate and ensures the robustness of our dataset.

• Page 11 last paragraph “nonspecific interactions introduced by the Y246K and C205A mutations, potentially leading to protein aggregation.“ DLS would be an effective measure of the aggregation states of each sample coming off the column.

While we appreciate the reviewer’s comment on the usefulness of DLS, we rule out the possibility of aggregation in subsequent experiments using DSF and native-PAGE (pages 12–13, lines 284–311). 

• Page 12, Catalytic activity of GGPPS mutants, first paragraph. Does not mention the wavelength at which malachite green dye absorbs.

The wavelength (620 nm) was already specified in the methods section (page 7, line 178).

• Page 12, last paragraph, “do not impair enzymatic activity”. The authors should rephrase this as they state under Catalytic activity of GGPPS mutants on page 12, “However, the DA mutant showed somewhat lower activity in this substrate concentration range,”

We carried out a statistical analysis of enzyme assay results to address Reviewer #2’s suggestion below (minor point 3), which showed that the slightly lower activity of the DA mutant was not significant. Therefore, we retained the phrase “do not impair enzymatic activity,” as it accurately reflects the findings. Instead, we rephrased the other sentence to: “The DA mutant showed slightly lower activity in this substrate concentration range, with an initial rate of 0.248 μM/min; however, this difference was not statistically significant” (pages 11–12, lines 258–262).

• Page 15 second paragraph second line “However, electron density data clearly indicated…”. Should include the sigma level used to make this determination.

The sigma level has been included in the updated manuscript as requested (page 15, lines 375–377).

• Page 18 Conclusion line 4 “For example, a dimeric form of GGPPS may provide a simpler system for investigating enzyme-inhibitor interactions. “ The dimeric form appears to be more easily crystallizable- but are we worried about introducing artefactual disulfide bonds (that typically do not exist in the wt) to mess with inhibitor binding?

We believe this study demonstrates that a dimeric form of GGPPS could serve as a useful tool for investigating inhibitor-binding interactions. However, this is a general statement and is not specifically referring to the DL mutant. We fully acknowledge the potential limitation of the DL mutant posed by the disulfide bonds, which may interfere with inhibitor binding during crystallization. This limitation was discussed in the results section (pages 17–18, lines 428–446), and in the conclusion, we specifically highlighted that ‘optimized’ dimeric forms of GGPPS could serve as valuable molecular tools (page 19, lines 463–465).

• Page 16 Conclusion line 10 “ While increased use of reducing agents during crystallization or further mutagenesis could address this issue,” The authors need to explore this more. Use different reductants (DTT?) to see if they can prevent the extra bonds from forming, so as to maintain an inhibitor binding environment similar to wt.

We fully agree with the reviewer that further exploration in this area would be helpful. Therefore, we have initiated studies employing strategies such as using an alternative reductant (TCEP), protecting cysteine residues with thiosulfate, and introducing additional mutations to replace cysteine residues. However, as the reviewer will appreciate as an expert crystallographer, crystallographic studies are time-intensive, and their outcomes are often unpredictable. Consequently, the results of these ongoing efforts, if successful, will be presented in a future communication.

• Figure 1A. Should mark the chains in the ribbon diagram. Then authors can refer to the same coloring scheme throughout the text.

We have labeled the subunit chains in Figure 1A as requested.

• Figure 5 the figure labels are unclear. The blue inset (A) shows some hydrogen bonds which are a little hard to see.

We have increased the font size of the labels to ensure clarity and consistency with the other figures. We have also modified the representation of the hydrogen bonds in the inset to improve their visibility (Figure 6 in the revised manuscript).

• Was the His-Tag cleaved?

The His-tag was not removed, and thus its removal was not discussed in the manuscript.

• Please describe the Imidazole gradient used for elution, starting and ending concentrations and number of column volumes, What is the composition of the elution buffer?

The elution details, including the imidazole gradient, column volumes, and the composition of the elution buffer, have been added to the methods section as requested (page 6, lines 156–160).

• What is the crystal morphology (a crystal picture would be helpful)? What were the crystal dimensions and how long it took for crystals to appear. Were the crystals stable over time, or did they dissolve over time?

The requested information about crystal morphology, dimensions, timeline, and stability has been added to the results section (page 15, lines 362–365). We have also included an example crystal image as a supplemental figure (S5 Fig).

• How long was the protein incubated with ligand for before setting up the trays?

The protein was incubated with ligands for 30 minutes prior to setting up the crystallization trays, as now detailed in the corresponding methods section (page 8, lines 198–201).

Reviewer #2

1. The authors state that “only two crystal structures of wild-type (WT) human GGPPS are available, one resolved at a moderate resolution of 2.7 Å, and neither in complex with a relevant inhibitor molecule”. I assume that the second structure is 6R4V, obtained at 2.2 Å resolution with a bound bisphosphonate (as attempted in this study). Therefore, this statement is unclear and seems to be misleading.

To address the reviewer’s concern about potentially misleading the readers, we have revised the statement as follows:

“Currently, only two crystal structures of wild-type (WT) human GGPPS are available, one resolved at 2.7 Å in complex with GGPP, the product of the enzyme, and the other at 2.2 Å in complex with ibandronate, a clinical bisphosphonate drug targeting the GGPPS paralogue, farnesyl pyrophosphate synthase [9, 10]. While these structures have provided valuable insights into the general architecture of the enzyme and its feedback inhibition mechanism, they do not capture interactions with more recently discovered nanomolar inhibitors, such as thienopyrimidine bisphosphonates.”

We believe this updated text (page 3, lines 71–77) provides greater clarity and accuracy, ensuring that the contributions of the previous structures are appropriately acknowledged. 

2. The mutants affect the catalytic activity, at least under the limited conditions inspected here. However, it is concluded that “the mutant proteins could be useful in functional studies, such as inhibitor screening, and aligns with the fact that the mutations are located on the protein surface, distant from the active site..”. I do not refute the possibility to use these constructs for inhibitor screening, but it seems that the active site is affected and this may have implications for drug discovery. Do the authors have any thoughts on how the mutations may affect catalytic activity and active site?

The mutation sites are indeed far from the active site, with the closest substrate-binding site located more than 15 Å away. This leads us to believe that the mutations do not directly affect active site residues. Interestingly, the mutations resulted in over a 30% increase in catalytic activity under sub-saturating substrate conditions, while no statistically significant changes were observed under saturating conditions. Given that the mutations are unlikely to directly influence the active site, we hypothesize that the observed changes in enzyme activity may be driven by a conformational mechanism. Determining the precise mechanism, however, would require a dedicated study involving detailed structural and kinetic analyses. At this stage, we can only propose a general statement about the potential involvement of a conformational mechanism. We have updated the paragraph in question to reflect this hypothesis (page 12, lines 270–282). 

3. TSA analysis – the change in Tm values is very much overinterpreted in my opinion. The change in Tm may be consistent with much of the claims, but does not prove most of them. For example, reduction in Tm is not a direct indication for change in the oligomeric state or conformational flexibility. Please reconsider the extent of interpretation in this section.

We fully agree with the reviewer that changes in the Tm values alone do not ‘prove’ alterations in the oligomeric state or conformational flexibility. This is why we employed multiple analytical methods to address these questions. Our conclusions were not solely based on TSA results but were supported by complementary data from SEC and native-PAGE. Upon close inspection, it should be evident that we approached these results with caution. For example, we deliberately used measured and intentional language that aligns with the reviewer’s comment: “Instead, their Tm values are ~1 °C lower than that of the DL mutant, effectively ruling out the possibility of higher-order oligomers and aggregates, a conclusion further supported by native-PAGE analysis (S3 Fig). The decrease in Tm values suggests slightly less stable structures for these mutants, consistent with increased conformational flexibility, as reflected in their size exclusion profiles.” (page 13, lines 307–311).

4. Similar to #2, the Kd for bisphosphonates is also altered to some extent by the mutations in the DSF studies. The entusiasm to use dimers in screening campaigns as suggested by the authors is less clear as the hexameric protein seems to behave well in solution and there are differences in affinity, so good inhibitors may be missed by using the mutant. Consider elaborating the pros and cons of this approach.

The use of dimer mutants in screening studies could offer the advantage of enhanced sensitivity in identifying initial hits, particularly during the early-stage screenings where only a few high concentrations of compounds are tested. For example, the basal Tm values of all three dimer mutants are ~10 °C lower than that of the WT protein, while the Tm changes induced by high concentrations of inhibitors are 2–3 °C greater for the DA and KL mutants compared to the WT protein. This increased ∆Tm makes the dimer mutants useful for detecting strong binders under these conditions. However, once hit compounds are identified, further characterization, including Kd determination and mechanistic studies, should be conducted using the WT protein to ensure an accurate assessment of inhibitor potency and selectivity. We have elaborated on these points in the manuscript as requested (page 14, lines 331–340).

5. The major problem is that the constructs were design to obtain structures of GGPPS bound to inhibitors, yet an inhibitor was not clearly identified despite the 2.1 Å resolution. With the 2.2 Å resolution structure of the hexameric enzyme bound to ibandronate in hand, the contribution of the designed dimer is unclear.

This work demonstrated that a simple protein engineering approach could potentially make GGPPS proteins more suitable for crystallographic analysis. It also showed that a non-active site single amino acid substitution can induce significant and unforeseen changes in protein structure and function. While these findings contribute to a broader understanding of protein engineering, one of the mutant constructs yielded the highest-resolution structure of GGPPS to date, providing the first characterization of IPP substrate binding. These contributions are clearly described in our manuscript, and we respectfully contend that they represent meaningful advancements deserving of publication and broad dissemination.

6. The lack of identified zolendronate also highlights the potential distortion of the active site by the mutation and resulting crystallographic artefacts (which obviously couldn’t have been predicted by alphafold…) which are deleterious for structural investigations aimed at improving small molecules binding. Did the authors try to fit a zolendroate molecule with partial occupancy?

As the reviewer rightly points out, crystallographic artefacts—specifically the inter-dimeric disulfide bonds—likely disrupted inhibitor binding during the crystallization of the DL mutant. We hypothesize that the disulfide bonds may interfere with the conformational changes required for tight ligand binding (pages 17–18, lines 429–440). Although we observed unexplained electron density at the expected inhibitor binding site, which could suggest partial occupancy of zoledronate, we decided not to model zoledronate into this density, as it was not sufficiently defined to confidently confirm ligand identity (page 18, lines 440–446).

Minor points:

1. Abstract – differential scanning, and not scattering, fluorimetry.

We have corrected the error as noted (page 1, line 32). 

2. Methods – did you mean 420 μM FPP and 300 μM IPP or perhaps 42 and 30, respectively? Otherwise I assume that the authors meant 100 times, and not 10 times, the Km for the saturating conditions.

We have updated the values from 420 and 300 to 42 and 30, respectively (page 7, line 172; page 11, lines 256–257). We sincerely thank the reviewer for their meticulous review, which brought this error to our attention.

3. Statistical analysis should be provided for differences in catalytic activity, etc.

Statistical analyses have been included for both catalytic activity and melting temperature determinations, as shown in Figures 3 and 4.

4. How did the authors made sure that the measured rates are initial? How was that defined?

The reaction time (5 minutes, as described in the methods section) was determined in preliminary assays to ensure the measurement of initial rates, during which the product concentration increased linearly with time. This also ensured that less than 10% of the substrate was converted to product by the time the reactions were terminated.

---

## [Decision Letter · Decision Letter 1]

30 Dec 2024

Engineering dimer mutants of human geranylgeranyl pyrophosphate synthase

PONE-D-24-41093R1

Dear Dr. Park,

We’re pleased to inform you that your manuscript has been judged scientifically suitable for publication and will be formally accepted for publication once it meets all outstanding technical requirements.

Kind regards,

Yoni Haitin, Ph.D.

Academic Editor

PLOS ONE

Additional Editor Comments (optional):

Reviewers' comments:

Reviewer's Responses to Questions

**Comments to the Author**

1. If the authors have adequately addressed your comments raised in a previous round of review and you feel that this manuscript is now acceptable for publication, you may indicate that here to bypass the “Comments to the Author” section, enter your conflict of interest statement in the “Confidential to Editor” section, and submit your "Accept" recommendation.

Reviewer #1: All comments have been addressed

Reviewer #2: All comments have been addressed

2. Is the manuscript technically sound, and do the data support the conclusions?

Reviewer #1: Yes

Reviewer #2: Yes

3. Has the statistical analysis been performed appropriately and rigorously? 

Reviewer #1: Yes

Reviewer #2: Yes

4. Have the authors made all data underlying the findings in their manuscript fully available?

Reviewer #1: Yes

Reviewer #2: Yes

5. Is the manuscript presented in an intelligible fashion and written in standard English?

Reviewer #1: Yes

Reviewer #2: Yes

6. Review Comments to the Author

Reviewer #1: All of my concerns have been addressed. I believe this manuscript is ready for publication. Thank you.

Reviewer #2: The authors have addressed all my concerns. While some points remain debated, I don't think that it should prevent the publication of this technically sound work.

7. PLOS authors have the option to publish the peer review history of their article (what does this mean?). If published, this will include your full peer review and any attached files.

Reviewer #1: **Yes: **Gloria Borgstahl

Reviewer #2: No

---

## [Editor Report · Acceptance letter]

3 Jan 2025

PONE-D-24-41093R1 

PLOS ONE

Dear Dr. Park, 

I'm pleased to inform you that your manuscript has been deemed suitable for publication in PLOS ONE. Congratulations! Your manuscript is now being handed over to our production team.

Kind regards, 

on behalf of

Dr. Yoni Haitin 

Academic Editor

PLOS ONE